# Fulminant Adenoviral-Induced Hepatitis in Immunosuppressed Patients

**DOI:** 10.3390/v14071459

**Published:** 2022-07-01

**Authors:** Juliane Kager, Jochen Schneider, Sebastian Rasch, Peter Herhaus, Mareike Verbeek, Carolin Mogler, Albert Heim, Gert Frösner, Dieter Hoffmann, Roland M. Schmid, Tobias Lahmer

**Affiliations:** 1Department of Internal Medicine II, School of Medicine, University Hospital Rechts der Isar, Technical University of Munich (TUM), 81675 Munich, Germany; juliane.kager@mri.tum.de (J.K.); jochen.schneider@mri.tum.de (J.S.); sebastian.rasch@mri.tum.de (S.R.); roland.schmid@mri.tum.de (R.M.S.); 2Department of Internal Medicine III, School of Medicine, University Hospital Rechts der Isar, Technical University of Munich (TUM), 81675 Munich, Germany; peter.herhaus@mri.tum.de (P.H.); mareike.verbeek@mri.tum.de (M.V.); 3Institute of Pathology, School of Medicine, Technical University of Munich (TUM), 80333 Munich, Germany; carolin.mogler@tum.de; 4German National Reference Laboratory for Adenoviruses, Institute for Virology, Hannover Medical School, 30625 Hannover, Germany; heim.albert@mh-hannover.de; 5Institute of Virology, School of Medicine, Technical University of Munich (TUM), 80333 Munich, Germany; gfrosner@gmx.de (G.F.); dieter.hoffmann@tum.de (D.H.)

**Keywords:** adenovirus, hepatitis, acute liver failure, rituximab, stem-cell transplantation

## Abstract

Human adenovirus (HAdV) can often lead to fulminant hepatitis in immunocompromised patients, mostly after reactivation of HAdV. Different risk factors, e.g., transplantation and chemotherapy, increase the risk of developing a HAdV hepatitis. We retrospectively analyzed three patients who showed the characteristics of a HAdV hepatitis observed in disseminated disease. In addition to PCR, diagnosis could be proven by pathology, CT scan, and markedly elevated transaminases. All patients had a hemato-oncologic underlying disease. Two had received a stem-cell transplant, and one was under chemotherapy including rituximab. Despite therapy with cidofovir, all patients died. As the incidence of HAdV hepatitis is low, diagnosis may be easily overlooked. No treatment approaches have yet been established. HAdV hepatitis should be considered as a differential diagnosis, especially when risk factors are present. To avoid dissemination, treatment should be initiated as soon as possible.

## 1. Introduction

Human adenovirus (HAdV)-induced hepatitis is a rare and, in most cases, fatal disease [1,2,3,4]. An infection with HAdV in immunocompetent patients leads typically to a mild disease, e.g., keratoconjunctivitis, respiratory tract infection, or gastrointestinal infection. In contrast, immunocompromised patients can suffer from systemic infections such as a severe pneumonia, colitis, or hepatitis [5,6]. HAdV infections are currently the focus of public attention following the increased incidence of severe hepatitis in children suspected to be caused by HAdV [7].

HAdV is a double-stranded DNA virus with different types being divided into seven species named A to G. Hepatitis is usually caused by species C [8,9]. Primary infection with HAdV occurs mostly during childhood with persistence of HAdV in lymphoid tissues such as tonsils and adenoids, as well as other sites [9,10,11,12,13]. Upon immunosuppression, a reactivation can possibly induce a fulminant infection [3,9,14]. Less frequently, an exogenous infection is the cause [15].

Within the population of immunosuppressed patients, there are various risk factors that may lead to a disseminated HAdV infection. These typically include stem-cell transplantation (SCT), solid-organ transplantation, human immunodeficiency virus (HIV) infection, or chemotherapy. In particular, post-allogenic SCT patients with graft-versus-host disease (GvHD), lymphopenia (absolute lymphocyte count <200/µL), use of anti-thymocyte globulin (ATG) or alemtuzumab therapy and T-cell-depleted grafts, or cord-blood administration are the most vulnerable populations [4,5,6,16]. Although these patients are at an increased risk, the diagnosis and the treatment of HAdV hepatitis are still challenging.

Here, we present three immunosuppressed patients with HAdV hepatitis who had different therapies, courses, and risk factors.

## 2. Case Presentation

### 2.1. Case 1

A 54 year old male patient was diagnosed with an IgG kappa-type multiple myeloma (MM). Treatment was started with conditioning chemotherapy followed by an autologous SCT. Due to progression of the MM, high-dose chemotherapy was performed, before an allogeneic SCT from a human leukocyte antigen (HLA) identical unrelated donor was conducted, which led to a complete remission. Because of the recurring transfusion requirement, a CD34^+^-selected stem-cell boost was applied that finally restored a normal full blood count. For this reason, the patient was on temporary immunosuppression with methotrexate (MTX) 2 months before HAdV viremia (Table 1).

After allogenic SCT, several complications occurred such as a GvHD of the skin (grade II) and the intestine (grade III-IV). While the GvHD of the skin could be controlled by therapy with prednisolone, the GvHD of the intestine was, for the most part, therapy-resistant. Various treatment attempts included prednisolone therapy, as well as ciclosporin, tacrolimus, everolimus, and mycophenolate mofetil (MMF). Moreover, the patient was treated by ruxolitinib, etanercept, human allogenic mesenchymal stromal cells, alpha antitrypsin 1, and an extracorporeal photopheresis. The symptoms were mostly controlled under the therapy of budesonide. 

In addition to the GvHD, the patient suffered from a transplantation-associated thrombotic microangiopathy, which was treated by eculizumab. To control an Epstein–Barr virus (EBV) reactivation, rituximab was administered three times, most recently 1 year before the HAdV infection occurred (Table 2).

The patient presented in our emergency room with fever. At this point, transaminases were markedly increased (maximum: SGPT 7984 U/L, SGOT 17460 U/L), while values had been normal 3 weeks before (Figure 1a). Gamma-glutamyl transferase and alkaline phosphatase were also elevated. Bilirubin was increased by a maximum of 2.5 mg/dL over time. In addition, the international normalized ratio was impaired, and elevated infectious parameters could be observed. Due to an acute liver failure (ALF) and cardiopulmonary instability, the patient was transferred to our intensive care unit (ICU).

Initially, we suspected a GvHD of the liver and administered 250 mg of prednisolone. Other differential diagnoses for viral hepatitis were excluded (Table 3); however, HAdV DNA was detected in blood by real-time polymerase chain reaction (PCR) (Table 4; for primer, see Appendix A). Genotyping in the reference lab revealed species C, type 5. In addition to HAdV hepatitis, the patient became symptomatic for pneumonia during inpatient stay and was respiratory insufficient when he was administered to the ICU. We, therefore, suspected a concomitant HAdV pneumonia, as bronchoalveolar lavage (BAL) fluid was also positive for HAdV. A computed tomography (CT) scan proved an atypical pneumonia showing global, reticular ground-glass opacities.

Transjugular biopsy of the liver revealed multifocal hepatocellular necrosis and a lobular chronic inflammation consistent with a viral hepatitis (Figure 2a). Histopathologically, a GvHD, as well as cytomegalovirus (CMV) and EBV infection, could be excluded. A CT scan presented an irregular contrasted parenchyma, a hepatic edema, and a dilatated common bile duct (Figure 3). Treatment with 325 mg of cidofovir and immunoglobulins was applied immediately after HAdV was diagnosed. The coagulation dysfunction was aggravated with recurrent bleeding complications. In addition, a central pulmonary embolism deteriorated the respiratory situation with development of treatment-resistant lactic acidosis caused by ALF. The patient died only 4 days after symptom onset due to fulminant multiorgan failure in disseminated HAdV infection.

### 2.2. Case 2

A 56 year old male patient was diagnosed with a diffuse large B-cell lymphoma (DLBCL). He was treated by six cycles of rituximab, cyclophosphamide, hydroxydaunorubicin, vincristine, and prednisolone (R-CHOP), followed by two cycles of rituximab. He developed a relapse of a composite lymphoma—histopathological and angioimmunoblastic T-cell lymphoma with an EBV-negative DLBCL. Therapy with rituximab, dexamethasone, cytarabine, and cisplatin (R-DHAP) was initiated (Table 1 and Table 2).

After the second cycle of R-DHAP, the patient presented in our emergency department due to fever. Although the patient had no subjective respiratory symptoms, he had a slight need for oxygen via nasal cannula. A CT image showed an atypical pneumonia with bipulmonary, left-accentuated ground-glass opacities. Broad-spectrum antibiotics were initiated.

During the inpatient stay, there was an acute increase in initially already elevated transaminases (maximum: SGPT 3194 U/L, SGOT 7459 U/L) (Figure 1b). Bilirubin peaked with 5.2 mg/dL over time. Moreover, the patient had an impaired hemostasis, a mild hepatic encephalopathy (HE), and hypoglycemia. The plasma disappearance rate of indo-cyanine green measured noninvasively by LiMON technology (Pulsion^®^ Medical Systems; Maquet Getinge Group Munich, Munich, Germany) was in between 2.6%/min and 4.8%/min (normal range 18–25%/min) [17]. This further confirmed the suspected reduced liver function. Because of the pronounced ALF, the patient was transferred to the ICU. A CT scan demonstrated an inhomogeneous contrasted liver parenchyma with dilatated bile ducts in the left lobe. The patient was treated with acetylcysteine, prednisolone (in total 550 mg in 3 days), and therapeutic plasma exchange until the result of the transjugular liver biopsy was available. It showed hepatic damage with extensive liver parenchymal necroses associated with an infectious pathogenesis (Figure 2b). While other differential diagnoses were ruled out virologically (Table 3), the infectious genesis turned out to be HAdV after PCR was positive in the blood (Table 4, for primer see Appendix A). HAdV was genotyped as species C; type 5. Therapy with 450 mg of cidofovir was initiated, accompanied by probenecid 5 days after symptom onset. In the case of fulminant hepatitis observed in disseminated disease, HAdV could also be detected in BAL fluid, and, as mentioned above, infiltrates were shown in a CT scan.

To bridge ALF, Advanced Organ Support (ADVOS multi, manufactured by Hepa Wash GmbH Munich, Munich, Germany)—an albumin dialysis system that detoxifies albumin bound metabolites—was initiated due to high ammonia levels and HE [18]. Moreover, we substituted immunoglobulins. The patient could be discharged with normalized transaminases (Figure 1b) and decreasing albeit still detectable HAdV PCR replicates. 

A weekly surveillance of HAdV viremia was performed, and immunoglobulins were adjusted to the number of copies presented in the PCR of blood samples. As chronic HAdV viremia was considered, chemotherapy was postponed until the patient had significant progress in his lymphoma. Balancing the risk due to the progress of his malignancy on one hand and the viremia on the other hand, chemotherapy was consequently continued.

The chosen treatment regime was rituximab, gemcitabine, and oxaliplatin (R-GemOx); according to a good tolerance, this was escalated to rituximab, ifosfamide, carboplatin, and etoposide (R-ICE). The use of corticosteroids was avoided to minimize the risk of HAdV reactivation.

Shortly before the initiation of chemotherapy, no viral load was detectable in PCR. Afterward, HAdV copies were mildly elevated. Therefore, a reduced dosage of 50 mg of cidofovir due to a kidney failure requiring dialysis was administered. Probenecid could not be applicated as the glomerular filtration rate was too low.

After chemotherapy, the patient developed neutropenic fever and eventually sepsis with respiratory insufficiency. Upon admission to ICU, the patient was asystole. Despite immediate resuscitation, the patient died 10 days after the last chemotherapy and 4 months after the first positive PCR for HAdV.

### 2.3. Case 3

A 40 year old female patient was diagnosed with acute myeloid leukemia (AML). After induction and consolidation chemotherapy, remission could be achieved, and an allogenic SCT from her haploidentical brother could be performed. Five months after SCT, a relapse of AML occurred. As salvage therapy did not lead to remission, a second allogenic SCT from her haploidentical sister was performed. Nevertheless, she suffered another extramedullary relapse manifesting itself as chloromas. Various approaches were taken to terminate progression such as irradiation of the multiple chloromas, sorafenib, and an anti-CD33/CD3 bispecific antibody. She also received four applications of donor lymphocyte infusion (DLI) from her sister and had treatment attempts including azacytidine, venetoclax, and gliteritinib. Most recently, a PET-CT showed multiple positive CXCR-4 spots. Therefore, radionuclide therapy against CXCR-4 and therapy with fludarabine, carmustine, melphalan, and ATG was initiated as a conditioning for her third allogenous SCT from a haploidentical unrelated donor (Table 1 and Table 2).

Five days after her third SCT, the patient presented with fever and tachycardia and was later transferred to the ICU and intubated due to septic shock and respiratory insufficiency. A PCR for HAdV was positive in blood and BAL samples (Table 4; for primer, see Appendix A) while a PCR for HAdV in stool had been negative 5 days before symptom onset. HAdV was genotyped as species C; type 5. Transaminases (maximum: SGPT 124 U/L, SGOT 191 U/L) and bilirubin increased (maximum: bilirubin 9.8 mg/dL) while coagulation deteriorated; a concomitant HAdV hepatitis was assumed (Figure 1c).

A CT scan showed an atypical pneumonia with bipulmonal consolidations, left basal accentuation, and ground-glass opacities matching the respiratory symptomatology, while the liver was without morphological abnormalities. A GvHD prophylaxis with MMF and ciclosporin was changed to MMF plus prednisolone, which we tapered. Due to leukopenia, we repeatedly applicated filgrastim. Therapy with cidofovir could not be initiated as the patient had acute kidney failure. Therapy with immunoglobulins was administered.

Given the fact that the patient’s underlying disease was not in remission and that she had multiorgan failure, it was decided in accordance with her relatives to not further escalate therapy. The patient died 1 month after symptom onset.

## 3. Discussion

Immunocompromised patients are at risk of developing HAdV viremia, resulting in fatal ALF. The three presented cases of HAdV hepatitis differed, showing the complexity of diagnosis and treatment.

Amongst other causes, ALF can be induced by different viruses. In particular, immunocompromised patients should be screened for hepatitis A virus, hepatitis B virus, hepatitis E virus, herpes simplex virus, varicella-zoster virus, EBV, and CMV [19] (Table 3). HAdV-induced ALF, however, can easily be missed as its incidence is low [1,20]. Most data are available for HAdV hepatitis following SCT; however, other immunosuppressants should also be taken into account. To the best of our knowledge, there are only two other cases published of HAdV hepatitis after rituximab therapy [4,21]. 

ALF itself is defined as an acute worsening of liver function parameters, coagulopathy, and an HE without pre-existing chronic liver disease (Table 5) [19]. In disseminated HAdV infections, both fulminant and milder forms of hepatitis can occur, as shown in case 3.

A PCR of the blood is considered the gold standard for diagnosing a HAdV viremia [5]. In addition to PCR, diagnosis of hepatitis with subsequent ALF can be confirmed by CT scan (Figure 3) and by liver biopsy (Figure 2), as shown in cases 2 and 3.

Case 1, case 2, and case 3 were typified as species C; whereas the clustered cases of hepatitis in childhood are suspected to be caused by F41 [22], species C is known to induce hepatitis in immunocompromised patients [9].

Patients with HAdV viremia typically present with fever, which does not simplify the diagnosis as it is a nonspecific symptom. Other symptoms may include lethargy, diarrhea, and jaundice [4]. Our three patients also had fever as an initial symptom (Table 5).

As incidence is low, there is no defined treatment regimen for a HAdV hepatitis [1,4,23]. Published guidelines focus on treatment recommendations for patients following SCT [5,6,9,16,23]. Accordingly, immunosuppression should be reduced as T-cell recovery contributes to viral clearance [4,5,8,9,16,23]. Patients 1 and 2 did not have immunosuppression when viremia evolved, or they were on local budesonide. They even underwent probationary therapy with prednisolone, which probably aggravated the hepatitis. When patient 3 developed symptoms 5 days after SCT, the risk of GvHD was high. We reduced immunosuppression as much as possible (Table 2).

In addition to reducing immunosuppression, most data are available for the antiviral agent cidofovir [5,6,16]. However, its use is controversial [9]. Cidofovir can inhibit viral DNA polymerase as a nucleotide analog and, therefore, reduce viral replication; however, application of cidofovir is limited by its nephrotoxicity [5,6,9]. This made discontinuation and reduced dosage in case 2 necessary, while, in case 3, it could not be initiated for this reason. Brincidofovir, a lipid conjugate of cidofovir, is proposed to be less nephrotoxic, but was not available in our hospital [9,24,25]. Unspecific T cells, DLI, which are intended to replace the missing immune response, are potent but also contain alloreactive T cells [6,9]. Therefore, a new therapeutic approach is the selection of HAdV-specific T cells, which seems promising; however, delayed and limited availability still confines its use [6,9,26]. We added immunoglobulins for therapy; however, there are no general recommendations and prospective data for their administration [4,5,27,28].

Mortality for HAdV hepatitis is estimated to be between 70% and 90% [1,4]. Despite the high mortality rate, case 3 did survive the acute phase of ALF. Moreover, we presented follow-up monitoring with adjunctive therapy using immunoglobulins in patient 3, which is unique as far as we know. Whether the neutropenic fever which led to case 3’s death was due to recurring HAdV replication or a different infection remains unclear.

In addition to HAdV hepatitis, it should be mentioned that all patients had a concomitant pneumonia presented as bipulmonal ground-glass opacities with consolidation (in case 3) in CT scans, which might have contributed to respiratory insufficiency. Moreover, PCR was positive in BAL fluid in all patients (Table 4).

Pathogenetically, it is considered that the replication of the virus begins in the gastrointestinal tract and further sheds to different tissues such as the bone marrow, lungs, liver, and pancreas, meaning that, in our cases, the lungs and especially the liver were the most important sites of replication [29]. However, given the rapid course of the disease and the lack of gastrointestinal symptoms in our cases, gastrointestinal involvement cannot be safely assumed. In general, a manifestation of HAdV in more than one site is associated with a poorer prognosis [20].

Because of its high mortality, regular PCR screening of blood or stool in post-allogenic SCT settings has been discussed, especially in patients with risk factors for HAdV viremia (Table 2), but not regularly performed [5,9,16,23]. For other patients at risk, no recommendations are available [5]. Interestingly, case 3 had a negative stool PCR 5 days before symptom onset/on the day of SCT, implying that there was no replication of HAdV in the intestine at that moment. Assuming that viremia precedes hepatitis, cases 1 and 2 already had elevated liver enzymes at the time of symptom onset [30] (Figure 1). No general recommendations for screening can be made, but it is hoped to hinder dissemination and organ manifestation.

To conclude, disseminated disease presented as fulminant hepatitis is a rare and, in most cases, lethal diagnosis. We could show very acute courses of the disease and present differential diagnoses and many diagnostics in detail. The diagnosis of HAdV hepatitis should be considered when typical risk factors are present. Ideally, viremia should be diagnosed before hepatopathy or other organ manifestation appears. Due to the high mortality rate, therapy should not be delayed and should be initiated as soon as possible. Further investigations are needed to establish a therapeutic algorithm and to identify further characteristics of this rare disease.

## Figures and Tables

**Figure 1 viruses-14-01459-f001:**
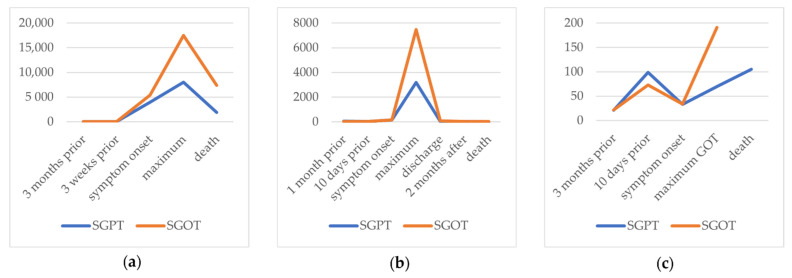
SGPT and SGOT (U/L) presented over time course for (**a**) case 1, (**b**) case 2, and (**c**) case 3.

**Figure 2 viruses-14-01459-f002:**
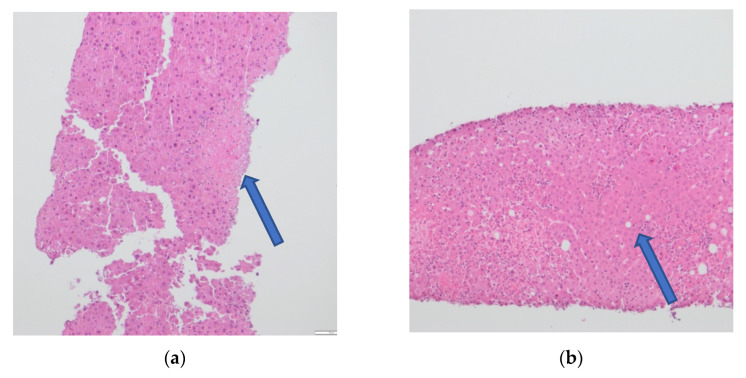
Liver biopsy (10× enlarged) in hematoxylin–eosin staining of (**a**) case 1 showing a multifocal, hepatocellular necrosis and viral inclusion bodies, and (**b**) case 2 showing an acute and extensive hepatocellular necrosis.

**Figure 3 viruses-14-01459-f003:**
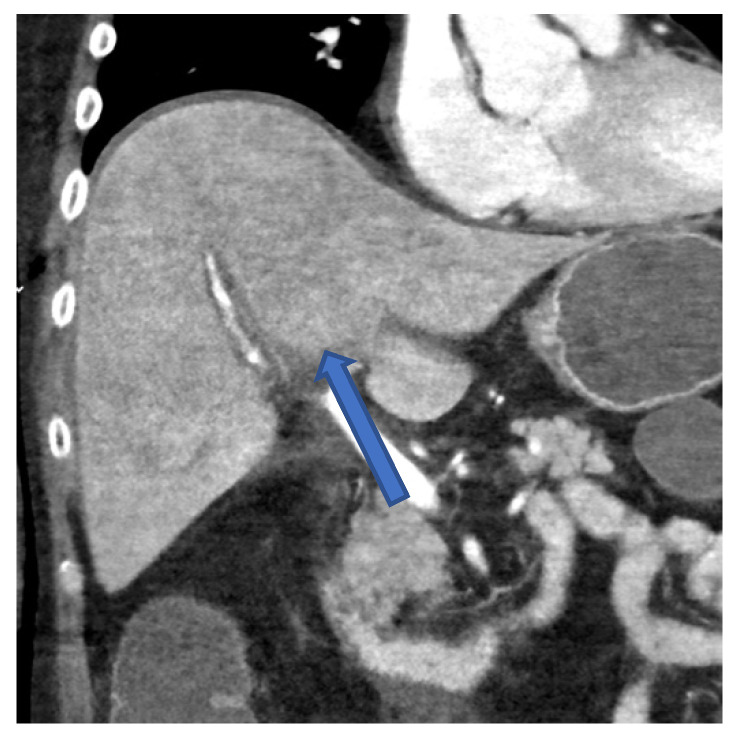
CT scan of the abdomen in frontal plane of case 1 showing an irregular contrasted liver parenchyma 2 days before the patient died.

**Table 1 viruses-14-01459-t001:** Patients’ characteristics.

	Sex	Age	Malignancy	SCT—Type
Case 1				1. autologous
	m	54	MM	2. allogenic—HLA identical, unrelated
				3. CD34+ selected stem cell boost—HLA identical, unrelated
Case 2	m	56	DLBCL	-
Case 3				1. allogenic—HLA identical, related
	f	40	AML	2. allogenic—HLA identical, related
				3. allogenic—HLA identical, unrelated

Abbreviations: HLA—human leukocyte antigen; MM—multiple myeloma; DLBC—Diffuse large B-cell lymphoma; AML—acute myeloid leukemia; SCT—stem-cell transplantation.

**Table 2 viruses-14-01459-t002:** Patients’ risk factors.

		Case 1	Case 2	Case 3
GvHD		Skin (grade II) Intestine (grade III–IV)	-	-
Immunosuppression (at/shortly before detection of viremia)		Budesonide (9 mg/day)		MMF (2 g/day) Ciclosporin
		Prednisolone (once 250 mg)	Prednisolone (550 mg in total)	
		MTX (1.5 months before)		
Rituximab (last application before symptom onset)		3 applications (12 months before)	10 + 2 applications (same month as symptom onset, 2 applications afterward)	-
SCT		Allogenic, unrelated	-	Allogenic, unrelated
Serotherapy	ATG	1 application	-	2 applications
	Alemtuzumab	-	-	-
T-cell-depleted grafts		CD34^+^-selected stem-cell boost	-	-
Lymphocyte count at symptom onset (per µL)		246	204	<100 leukocytes

Abbreviations: ATG—anti-thymocyte globulin; GvHD—graft-versus-host disease; MMF—mycophenolate mofetil; MTX—methotrexate; SCT—stem-cell transplantation.

**Table 3 viruses-14-01459-t003:** Hepatitis—viral differential diagnoses.

		Case 1	Case 2	Case 3
HAV, HBV, HCV, HEV	Serology	HAV IgG, anti-HBs positive, the further negative	HAV IgG, anti-HBs positive, the further negative	-
HEV	PCR (stool/blood)	Negative (stool)	Negative (blood)	-
HSV	PCR (blood)	Little amount	Negative	-
EBV	PCR (blood)	Negative	Negative	Negative
CMV	PCR (blood)	Negative	Negative	Negative
HHV-6	PCR (blood)	Negative	-	-
HHV-7	PCR (blood)	Little amount	-	-
HIV	ELISA (blood)	Negative	Negative	-
Enterovirus	PCR (blood)	Negative	-	-
Parovirus B19	PCR (blood)	Little amount	-	-
VZV	PCR (blood)	Negative	Negative	-

Abbreviations: EBV—Epstein–Barr virus; ELISA—enzyme-linked Immunosorbent Assay; CMV—cytomegalovirus; HAV—hepatitis A virus; HBV—hepatitis B virus; HCV—hepatitis C virus; HEV—hepatitis E virus; HHV—human herpesvirus; HIV—human immunodeficiency viruses; HSV—herpes simplex virus; PCR—polymerase chain reaction; VZV—varicella-zoster virus.

**Table 4 viruses-14-01459-t004:** Virological diagnostics.

		Case 1	Case 2	Case 3
Viral load PCR (maximum Geq/mL)	Blood	5.4 × 10^10^	1.1 × 10^9^	1.0 × 10^8^
	BAL	1.3 × 10^7^	1.3 × 10^7^	3.8 × 10^5^
Regular screening for HAdV	Blood Stool	no no	no no	no no
HAdV serology		IgG-negative, IgA-negative	IgG-positive, IgA-negative	-

Abbreviations: BAL—bronchoalveolar lavage; HAdV—human adenovirus; PCR—polymerase chain reaction.

**Table 5 viruses-14-01459-t005:** Clinical course.

		Case 1	Case 2	Case 3
Initial symptom		Fever	Fever	Fever, tachycardia
Acute liver failure	Acute rise of transaminases	Yes	Yes	Mild
	HE	Unknown	Yes	No
	Coagulopathy	Yes	Yes	Yes
	Chronic liver disease	No	No	No
HAdV pneumonia		Yes	Yes	Yes
Death after symptom onset (in days)		4	116	27

Abbreviations: ALF—acute liver failure; HAdV—human adenovirus; HE—hepatic encephalopathy.

## Data Availability

Not applicable.

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
