# Peer review of "Fulminant Adenoviral-Induced Hepatitis in Immunosuppressed Patients"

_viruses, 2022, doi:10.3390/v14071459_

Round 1
Reviewer 1 Report
This study provides three cases of fulminate HAdV hepatitis in immunosuppressed patients. These data are a good complement to the existing evidence.
The main remarks are the following:
- The authors mentioned that liver tissue has revealed histopathologic evidence of a viral hepatitis, but it is not clear whether the detection of HAdV in the liver tissue was confirmed, and by what method, for example immunohistochemistry, in situ hybridization, or culture? Those are necessary to the confirmation of HAdV hepatitis.
- HAdV DNA was detected in blood by PCR in all three patients, and it was quite interesting that BAL fluid also showed positive results. However, the details of respiratory symptoms were not stated, did all patients showed clinical symptoms of respiratory tract infections? More importantly, the subtype of HAdV was also not specified.
- The discussion part could be improved, as there have been some reports on HAdV hepatitis in immunosuppressed patients, the author need to emphasize more on what is unique about thesethree cases and what does it add to the literature.
- The ages of case1 and case2 were inconsistent in the text and Table1.
Author Response
Dear editor, dear reviewers,
Please find attached a revised version of our manuscript. Your comments and those of the reviewers were highly insightful and enabled us to greatly improve the quality of our manuscript. In the following pages are our point-by-point responses to each of the comments of the reviewers as well as your own comments. The language has been checked for correctness by native speakers.
Revisions in the text are shown using “tracked changes”. Highlighted text for additions [example], and strikethrough font [example] for deletions. We hope that the revisions in the manuscript and our accompanying responses will be sufficient to make our manuscript suitable for publication in Viruses.
With kind regards,
PD Dr. med. Tobias Lahmer
Klinik und Poliklinik für Innere Medizin II, Klinikum rechts der Isar der Technischen Universität München
Ismaninger Str. 22,
81675 Munich
Telefon: +49 89 4140 9345
Fax: +49 89 4140 6243
Email: TobiasLahmer@me.com
Dear Reviewer 1,
we very appreciate your careful reading and your recommendations. We would like to consider your suggestions for improvement:
- The authors mentioned that liver tissue has revealed histopathologic evidence of a viral hepatitis, but it is not clear whether the detection of HAdV in the liver tissue was confirmed, and by what method, for example immunohistochemistry, in situ hybridization, or culture?
Those are necessary to the confirmation of HAdV hepatitis.
The histomorphological presentations revealed a virological genesis of the hepatitis. In our opinion, the diagnosis is in the presence of clinical characteristics and positive PCR for HAdV unambiguous. Since confirmation takes several weeks and to not delay therapy, we decided not to perform further specification in the interest of the patients. We tried to make it clearer in the manuscript.
- HAdV DNA was detected in blood by PCR in all three patients, and it was quite interesting that BAL fluid also showed positive results. However, the details of respiratory symptoms were not stated, did all patients showed clinical symptoms of respiratory tract infections?
We agree that the accompanying pneumonia should also be described in detail. We therefore added details in concern of respiratory symptoms to each case presentation.
- More importantly, the subtype of HAdV was also not specified.
We performed typing and results are shown in the manuscript.
- The discussion part could be improved, as there have been some reports on HAdV hepatitis in immunosuppressed patients, the author need to emphasize more on what is unique about thesethree cases and what does it add to the literature.
Since there are only few described cases for HAdV hepatitis we think that every presented case is of concern. Besides, we feature many differential diagnoses and wide spectrum of diagnostics as CT scan, virological assessments and liver biopsies. Case 2 takes a special role as this patient did not receive a SCT while in literature most cases described had a SCT. Further, this case did survive the acute phase of the ALF. In the meantime, the patient has died which can either be attributed to neutropenic fever with a common infection or a recurrent replication of HAdV. We also present a follow-up of this patient.
- The ages of case 1 and case 2 were inconsistent in the text and Table1.
Thank you for checking the paper so intensively. We corrected this error.

Reviewer 2 Report
Nice collection of adenovirus hepatitis cases, well presented but with a few shortcomings.
important points:
The authors do neither mention nor consider "disseminated disease" caused by HAdV in HSCT recipients. Actually their patients fulfill several diagnostic criteria for disseminated disease, for example
- "sepsis like symptoms" (e. g. fever)
- high virus loads in blood
- symptomatic infection of more than one organ (liver and lung)
Therfore, their patients (at least #1 and #2) should rather be labeled as "fulminant hepatitis observed in dissemianted disease" or "disseminated disease presenting as fulminant hepatitis" or "fulminant hepatitis as main symptom of disseminated disease".There was also a case report on fulminant hepatitis in a patient with disseminated disease which could be cited (DOI 10.1002/jmv.21071).
In patient #3 I am not so sure about "fulminant" hepatitis (because of the low transaminase peaks) but in my view this is a highly interesting case because it shows adenovirus disseminated disease in a patient without HSCT!
Another shortcoming is that adenovirus was not typed in these three cases. If typing results are available, these should be presented in the manuscript. If typing was not performed, it should be done if still feasible. The point "typing" is currently very important because of the epidemic of hepatitis cases in children with a questionable link to HAdV-F41 infections.
minor point:
the term "subtypes" (line 37) is wrong, Taxonomically correct is the term "types"
Author Response
Dear editor, dear reviewers,
Please find attached a revised version of our manuscript. Your comments and those of the reviewers were highly insightful and enabled us to greatly improve the quality of our manuscript. In the following pages are our point-by-point responses to each of the comments of the reviewers as well as your own comments. The language has been checked for correctness by native speakers.
Revisions in the text are shown using “tracked changes”. Highlighted text for additions [example], and strikethrough font [example] for deletions. We hope that the revisions in the manuscript and our accompanying responses will be sufficient to make our manuscript suitable for publication in Viruses.
With kind regards,
PD Dr. med. Tobias Lahmer
Klinik und Poliklinik für Innere Medizin II, Klinikum rechts der Isar der Technischen Universität München
Ismaninger Str. 22,
81675 Munich
Telefon: +49 89 4140 9345
Fax: +49 89 4140 6243
Email: TobiasLahmer@me.com
Dear Reviewer 2,
thank you for improving our manuscript with your valuable comments. We will answer these point by point:
- The authors do neither mention nor consider "disseminated disease" caused by HAdV in HSCT recipients. Actually their patients fulfill several diagnostic criteria for disseminated disease, for example
- "sepsis like symptoms" (e. g. fever)
- high virus loads in blood
- symptomatic infection of more than one organ (liver and lung)
Therefore, their patients (at least #1 and #2) should rather be labeled as "fulminant hepatitis observed in dissemianted disease" or "disseminated disease presenting as fulminant hepatitis" or "fulminant hepatitis as main symptom of disseminated disease". There was also a case report on fulminant hepatitis in a patient with disseminated disease which could be cited (DOI 10.1002/jmv.21071).
Thank you for pointing out this relevant information. You are right that it is important to elaborate on the dissemination and the paper you attached has underlined this. We, therefore, added information on dissemination and included the mentioned paper into our references.
In patient #3 I am not so sure about "fulminant" hepatitis (because of the low transaminase peaks) but in my view this is a highly interesting case because it shows adenovirus disseminated disease in a patient without HSCT!
We agree that case 3 had a hepatitis although not directly fulminant. The course of disease in general, however, can be described as fulminant. We agree that case 2 is an important case as this patient did not receive SCT what we emphasized once again. It was important for us to show the spectrum of the disease – patient 1 with SCT and fulminant hepatitis, patient 2 without SCT but still fulminant hepatitis, patient 3 with SCT and fulminant course of the disease although hepatitis was not clearly fulminant.
- Another shortcoming is that adenovirus was not typed in these three cases. If typing results are available, these should be presented in the manuscript. If typing was not performed, it should be done if still feasible. The point "typing" is currently very important because of the epidemic of hepatitis cases in children with a questionable link to HAdV-F41 infections.
We could perform and add typing. Results are added to the manuscript.
- The term "subtypes" (line 37) is wrong, Taxonomically correct is the term "types".
Thank you for reading with attention, we corrected this error.

Reviewer 3 Report
The review of manuscript
Title: Fulminant adenoviral induced hepatitis in immunosuppressed patients
- Summary
In this manuscript, the authors reported three cases which is fulminant adenoviral induced hepatitis. It is valuable and critical to report rare cases related with adenovirus induced hepatitis. In this manuscript, the authors well-organized data to report the cases and well-written the manuscript. However, the manuscript has to carefully consider reviewing next minor questions below.
- Major issues
- What is the serotype or genotype of HAdVs found in three patients? It is important to identify the serotype or genotype of HAdVs. If the authors have no antibody for serotype testing, simply the authors can perform sequencing on Penton base, Hexon and Fiber genes. From the results, the genotype of HAdV can be identified.
- Was the regular screening for HAdV the PCR from stool samples? The authors did not find HAdV from stool sample in case 3, but found HAdV in blood sample. The authors mentioned PCR in blood sample is the gold standard for diagnosing a HAdV viremia. Are there any reasons to show different, critical result from the two different sampling methods?
- Show all primers used in this study by a table.
Author Response
Dear editor, dear reviewers,
Please find attached a revised version of our manuscript. Your comments and those of the reviewers were highly insightful and enabled us to greatly improve the quality of our manuscript. In the following pages are our point-by-point responses to each of the comments of the reviewers as well as your own comments. The language has been checked for correctness by native speakers.
Revisions in the text are shown using “tracked changes”. Highlighted text for additions [example], and strikethrough font [example] for deletions. We hope that the revisions in the manuscript and our accompanying responses will be sufficient to make our manuscript suitable for publication in Viruses.
With kind regards,
PD Dr. med. Tobias Lahmer
Klinik und Poliklinik für Innere Medizin II, Klinikum rechts der Isar der Technischen Universität München
Ismaninger Str. 22,
81675 Munich
Telefon: +49 89 4140 9345
Fax: +49 89 4140 6243
Email: TobiasLahmer@me.com
Dear Reviewer 3,
thank you for considering our manuscript and for adding important facts. We are happy to address your alterations.
- What is the serotype or genotype of HAdVs found in three patients? It is important to identify the serotype or genotype of HAdVs. If the authors have no antibody for serotype testing, simply the authors can perform sequencing on Penton base, Hexon and Fiber genes. From the results, the genotype of HAdV can be identified
It was possible for us to genotype HAdV and the results are demonstrated in the manuscript.
- Was the regular screening for HAdV the PCR from stool samples?
There was neither regular screening in blood nor regular screening in stool. We added this information to Table 4.
- The authors did not find HAdV from stool sample in case 3 but found HAdV in blood sample. The authors mentioned PCR in blood sample is the gold standard for diagnosing a HAdV viremia. Are there any reasons to show different, critical result from the two different sampling methods?
As replication is considered to begin in the gastrointestinal tract, it is discussed if PCR in stool is positive before PCR in blood but general recommendations in concern of screening methods are lacking. We added further details to understand the pathogenesis.
- Show all primers used in this study by a table.
The primers are added to Table S1.
